# Modulating Gastrointestinal Microbiota in Preweaning Dairy Calves: Dose-Dependent Effects of Milk-Based Sodium Butyrate Supplementation

**DOI:** 10.3390/microorganisms12020333

**Published:** 2024-02-05

**Authors:** Donglin Wu, Zhanhe Zhang, Qifan Song, Yang Jia, Jingwei Qi, Ming Xu

**Affiliations:** 1College of Animal Science, Inner Mongolia Agricultural University, Hohhot 010018, China; wdl2017@emails.imau.edu.cn (D.W.); nddky@emails.imau.edu.cn (Z.Z.); jiayang@imau.edu.cn (Y.J.);; 2National Center of Technology Innovation for Dairy, Hohhot 010080, China

**Keywords:** calf feeding, sodium butyrate, milk, ruminal and intestinal microbiota, cross-talk

## Abstract

Sodium butyrate (SB), an essential nutritional additive for livestock, has drawn notable interest for its potential for enhancing microbiota development in ruminant animals. This study aimed to assess SB’s effects on ruminal and intestinal microbiota when added to milk for preweaning dairy calves nearing 45 days old. We administered SB in the calves’ milk at four levels: 0 g/d (control), 4.4 g/d (low), 8.8 g/d (medium), and 17.6 g/d (high). After a six-week trial with ten replicates per group, ruminal fluid and fecal samples were collected for 16S rRNA sequencing, specifically targeting the V3–V4 regions to analyze microbiota. The results indicated an enhancement in ruminal microbiota, particularly in community richness, with low-level SB supplementation but minimal benefits from medium and high levels of supplementation. Increasing the level of SB supplementation had a negative impact on intestinal microbiota, affecting community richness and some potentially beneficial bacterial genera. However, low SB supplementation could positively adjust the communication between ruminal and intestinal microbiota. Overall, this study suggests feeding milk supplemented with a low level of SB to suckling calves close to an older age to promote ruminal microbiota development.

## 1. Introduction

The early nurturing of dairy calves is crucial, setting the stage for their critical role as efficient milk producers in adulthood and necessitating proper care during the preweaning phase to optimally prepare them for future lactation success [1]. The gastrointestinal microbiota, a vast and distinct genomic entity, is pivotal to the health and growth of calves, drawing heightened research focus for its integral role in the well-being of these young animals [2,3].

Butyrate, a by-product of anaerobic microbial fermentation on the dietary canvas, stands out as a nutritional powerhouse, taking center stage as an asset for reinforcing the health and progression of young animals [4]. In the realm of nutrition and metabolism, butyrate stands out, as it crucial for the nascent stages of nutrient assimilation in calves. It revolutionizes the landscape of energy use in the body, offering an abundantly efficient fuel source for the digestive system’s cells, from the rumen to the colon [4,5]. Delving into the domain of gut health and structural integrity, we can point out that butyrate emerges as a cornerstone, safeguarding mucosal defenses, subduing inflammation, and nurturing the vital processes of cellular renewal along the intestinal lining [6]. When considering immunomodulatory influences, butyrate casts its salutary effects far and wide, deftly improving local gut immunity alongside the broader systemic defense network for optimal calf well-being [4,7]. In relation to disease resistance, this compound’s role is undeniable. Butyrate serves as a deterrent to pathogenic siege within colonic epithelial cells while simultaneously strengthening the young calf’s resilience to persistent viral and bacterial threats [8,9]. Turning to its origins and interplay within the gut microbiota, we can point out that butyrate establishes its prominence among the short-chain fatty acids (SCFAs), which are essential players in nurturing a balanced microbial habitat within the gut [10]. Furthermore, there is growing and burgeoning agreement among experts that butyrate has potential in expediting the developmental maturity of the intestinal microbiota in dairy calves [11]. Together, these insights reveal the intricate tapestry of butyrate’s role in calf physiology, illustrating its profound alignment with vital biochemical corridors.

Supplementing exogenous butyrate in liquid feed for preweaning calves appears to be a reasonable and effective method for ensuring butyrate intake during feeding [11]. It has been reported that milk replacers introducing exogenous butyrate—such as sodium butyrate (SB), encapsulated variants of SB, and tributyrin [9,12,13]—significantly impact the population of key SCFA-producing bacteria, as well as bacteria associated with health in the hindgut. This influence can even extend to the rumen microbiota [9]. However, the subject of butyrate supplementation in milk and its impact on the development of gastrointestinal microbiota is less studied. It has been identified that pasteurized waste whole milk supplemented with tributyrin during periods close to weaning has effects on calf health [14], but the impacts on the ruminal microbiota remain undetermined. Our antecedent study, which employed a low to high gradient of SB additives in newborn (day 17) dairy calves, found that an appropriate dosage of SB supplementation increased the development of intestinal microbiota [11]. However, it remains unclear whether this positive outcome extends to prolonged supplementation with SB and whether it influences the ruminal and intestinal microbiota. Thus, further research is required to affirmatively understand such effects.

During the period following the newborn stage, a higher quantity of exogenous butyrate is necessary, as its production from immature microbiota fermentation is low [4,11,15]. Our prior study concluded that the optimal dose of exogenous SB supplementation was reduced when analyzing the host’s growth phenotype (weekly average daily gain in body weight) in calves [11]. Consequently, based on these previous findings, there is a growing interest in examining the gastrointestinal microbiota in conjunction with extended periods of SB supplementation. With this in mind, the goals of the present study are to evaluate how exogenous SB supplementation influences the rumen and intestinal microbiota at a later developmental stage (day 45) in dairy calves that are fed milk. This study aims to expand our understanding of the dynamics between SB supplementation and the development of the gastrointestinal microbiota over an extended period and in older calves.

## 2. Materials and Methods

### 2.1. Study Subjects and Research Design

The study protocol underwent review by and received approval from the Institutional Animal Care and Use Committee at Inner Mongolia Agricultural University, with the approval number being 2020099.

The current study’s experimental design was consistent with our recently published study [11]. In brief, eighty robust Holstein calves, including 56 females and 24 males, with an average body weight (BW) of 41.72 ± 2.81 kg and aged 2 to 4 days (2.88 ± 0.45 days), were randomly distributed into four treatment groups (each group consisting of 20 calves with 14 females and 6 males) according to their age and BW. The treatments involved feeding calves milk with varying amounts of SB supplementation for calves: (1) control without SB, CON; (2) low level of SB, LSB; (3) medium level of SB, MSB; and (4) high level of SB, HSB. The daily supplementation of SB for the LSB, MSB, and HSB groups was established at 4.4 g, 8.8 g, and 17.6 g, respectively, equally divided across two feedings, resulting in per-feeding quantities of 2.2 g for LSB, 4.4 g for MSB, and 8.8 g for HSB. The highest level of SB supplementation was set at 17.6 g/d, referenced from prior groundwork established in a pre-experiment and documented by Wu et al., 2023 [11], where we saw no further promotive effects on preweaning calves at levels exceeding this concentration in milk. Hence, 17.6 g/d was selected as the ceiling. To comprehensively explore the effects of SB supplementation on calf growth, we established two intermediary dosages between the zero-supplementation baseline (0 g/d) and the peak (17.6 g/d). These were 4.4 g/d, representing the low level, and 8.8 g/d, representing the moderate level, facilitating an evaluation of the incremental effects across distinct SB concentrations in our study. The SB product (brand Jiabaoyu, with purity of ≥98% SB; produced by Jinan Degao Agriculture and Animal Husbandry Technology Co., Ltd., Jinan, China) utilized in this study took the form of a raw powder. For SB supplementation, SB was manually mixed into individual open buckets for each calf in the SB treatment groups before each feeding.

The study was carried out at the Shazhou Dairy Company located in Liangcheng County within Ulanqab, China, pinpointed at a latitude of 40.53° N and a longitude of 112.49° E. The calves involved in our study were sourced exclusively from this facility. This region experiences a mid-temperate continental monsoon climate with an average humidity of 30.78 ± 30.67% and an average temperature of 18.42 ± 6.85 °C, where are both reported as means plus or minus standard deviation. To monitor relative humidity and ambient temperature, readings were taken three times daily: early morning from 6 to 8 a.m., midday between 1 and 3 p.m., and in the evening from 6 to 8 p.m. We collected these climate data with the help of Hobo Pro Series Temp probes, supplied by Onset Computer Co. based in Pocasset, MA, USA. These instruments, which were suspended roughly 1.0 m off the ground, logged environmental conditions at 10 min increments throughout the study.

Calves were promptly separated from their multiparous Holstein mothers, who had experienced 3–4 deliveries, signifying a stable genetic line. These newborns were individually housed in well-ventilated outdoor hutches (the dimensions are 1.35 m tall, 1.5 m wide, and 2.2 m deep) equipped with absorbent dry sand bedding, a setup that facilitates cleanliness and comfort. Every morning at 10 A.M., the bedding was meticulously cleaned to ensure sanitary conditions. To establish baseline health, each calf underwent a thorough clinical examination before any experimental protocols commenced. Throughout the study, their health and development were vigilantly monitored to ensure the integrity of the experiment. To emulate natural feeding patterns and bolster immune defenses, the calves received an amount of pasteurized colostrum equivalent to 10% of their birth weight within the initial 60 min of life, followed by a subsequent 2 L after a 6 h interval, thus ensuring a robust start to life.

Normal, unpasteurized milk from the farm’s supply was processed and pasteurized to feed the calves, ensuring both safety and nutrient retention. Individual feeding buckets, designated for each calf, provided a controlled and hygienic feeding environment. These buckets underwent thorough cleaning and drying post-feeding to prevent bacterial growth. A structured feeding regimen featured two daily portions of pasteurized milk—4.4 L from the fourth to the tenth day and doubling that amount to 8.8 L from the eleventh to the forty-fifth day. These feedings, administered at 05:00 and 17:00, were carefully temperature-regulated between 36 and 37 °C to closely mimic the natural warmth of milk, which facilitates digestion and acceptance by the calves. During this critical developmental phase leading up to weaning, calves had unrestricted access to a specialized commercial pelleted starter feed and water, served in distinct containers to foster independent eating and drinking habits. To simplify the diet and focus on essential nutrients, hay was excluded from their dietary regimen.

The pelleted starter, sourced from Tianjin Jiuzhou Dadi Feed Co., Ltd. (Tianjin, China), was introduced on the third day. This early introduction promotes early rumen development and prepares the calves for future feedstuffs. The study by Wu et al. (2023) [11] presents a trove of information outlining specific feeding tactics, management practices, and the compositional analysis of the utilized starter feed, granting full insight into the nutritional strategy employed.

### 2.2. Sample Collection

On the concluding day of the experimental period, marked by day 45, we repeated the sampling protocol implemented on day 17 to procure gastrointestinal microbiota specimens from the selected calf cohort. The total cohort of forty calves was divided into evenly aged and weighted groups of ten for sampling—each group comprising seven female and three male calves. As outlined by Wu et al. (2023) [11], precautions were taken to collect fecal matter directly from the rectum for intestinal microbiota analysis. This process occurred precisely 4 h following the calves’ morning milk intake. Exercising stringent sterility measures, the sample collection was performed with sterile gloves, ensuring that environmental exposure was meticulously avoided to prevent external microbial interference. Once acquired, the fecal samples were placed into sterile, pyrogen-free centrifugal tubes. The samples were then immediately preserved with rapid freezing in liquid nitrogen and subsequently sequestered at a chilly −80 °C for future examinations.

The method for extracting rumen fluid samples, geared toward ruminal microbiota analysis, was carried out as per a method documented in the literature [16]. Harvesting occurred from the oral cavity, similarly timed at 4 h after the morning milk ingestion. This entailed the use of a supple esophageal probe (A1164K; Anscitech Co., Ltd., Wuhan, China) specifically designed for such a purpose. We began by discarding the initial 10 mL draw to obviate the possibility of salivary adulteration. Following this step, we collected 10 to 20 mL of the rumen liquid, which was promptly allocated into sterile, endotoxin-free centrifugal tubes. In alignment with the fecal specimen protocol, these liquid samples were also rapidly frozen using liquid nitrogen and preserved at −80 °C to await rigorous analysis.

### 2.3. Sample Analysis

Microbiota analysis of the ruminal and fecal samples required meticulous DNA extraction, PCR amplification, and 16S rRNA gene sequencing. This comprehensive process has previously been outlined by Wu et al. (2023) [11].

For DNA extraction, we obtained total DNA from ruminal and fecal specimens utilizing the hexadecyl trimethyl ammonium bromide (CTAB) method and the E.Z.N.A.^®^ Soil DNA Kit (Omega Bio-Tek, Norcross, GA, USA) following the manufacturer’s protocol and validation procedures detailed previously [17]. We assessed the DNA integrity with 1% agarose gel electrophoresis and quantified the concentration and purity via spectrophotometry using NanoDrop 2000 (Thermo Scientific, Waltham, MA, USA).

The PCR amplification and subsequent 16S rRNA sequencing entailed targeting the V3–V4 hypervariable regions using primers 338F (5′-ACTCCTACGGGAGGCAGCA-3′) and 806R (5′-GGACTACHVGGGTWTCTAAT-3′) on the Illumina MiSeq platform (Majorbio BioPharm Technology, Shanghai, China). A rigorous thermal cycling protocol was employed: an initial denaturation at 95 °C for 3 min, followed by 29 cycles of denaturation at 95 °C for 30 s, annealing at 55 °C for 30 s, extension at 72 °C for 45 s, and a final extension at 72 °C for 10 min. Each 20 μL PCR reaction contained a mixed composition: PrimeSTAR buffer, dNTPs, forward and reverse primers, PrimeSTAR HS DNA Polymerase (Takara Bio, Mountain View, CA, USA), and the template DNA. Post-amplification, 2% agarose gels verified successful amplification, and PCR products were purified using Axygen Biosciences’ DNA purification kit (Union City, CA, USA).

Sequencing data processing included demultiplexing and stringent quality filtering with fastp version 0.20.0 (HaploX Biotechnology, Shenzhen, China), followed by merging using FLASH version 1.2.7 (https://ccb.jhu.edu/software/FLASH/, accessed on 15 March 2022) based on protocols established previously [5]. UPARSE (version 7.0) (http://drive5.com/uparse/, accessed on 15 March 2022) [18] grouped Operational Taxonomic Units (OTUs) with 97% similarity, with chimeric sequences identified and removed according to the approach by Chen et al. (2018) [19]. Taxonomic classification of each OTU was determined through the representative sequence analysis employing the RDP Classifier algorithm (http://rdp.cme.msu.edu/, accessed on 12 November 2021) benchmarked against the Silva 16S rRNA SSU123 database (https://www.arb-silva.de/, accessed on 12 November 2021) with a 70% confidence threshold [20].

### 2.4. Microbial Data Analysis and Statistical Analysis

Alpha diversity was assessed using various indices such as community diversity (Shannon index, Simpson index, and phylogenetic diversity (Pd) index), community richness (Sobs index, Ace index, and Chao1 index), and community evenness (Shannoneven index and Simpsoneven index). For beta diversity (diversity between samples), the weighted UniFrac distance was used, followed by Adonis analysis, and visualization was performed using nonmetric multidimensional scaling (NMDS). Adonis is a nonparametric technique that tests differences in community structures among populations. Unique and core microorganisms at the phylum, genus, and OTU levels were depicted using Venn diagrams. Dominant bacteria at both the phylum and genus classifications were identified as those representing a prevalence of ≥0.1% in all groups examined. These core bacterial populations were subsequently subjected to rigorous statistical analysis. Regarding the presence of nonspecific taxonomic ranks, terms such as ‘unclassified’ and ‘norank’ were removed from the results section. The relationship between ruminal microbiota and intestinal microbiota in each group was analyzed using a Two-Matrix Correlation Heatmap (Pearson) on the Majorbio Cloud Platform (https://cloud.majorbio.com/page/tools/, accessed on 15 April 2023).

Before statistical analysis, data were checked for homogeneity of variance and transformed as needed. Microbiota-related indices (alpha diversity index, taxonomic analysis at phylum and genus levels) were analyzed with one-way ANOVA using SPSS statistical software (version 24.0; SPSS Inc., Chicago, IL, USA). We then refined mean differences using the Tukey–Kramer post hoc test. To evaluate the impact of treatment levels (specifically, the degrees of SB supplementation), we utilized both linear and quadratic orthogonal polynomial comparisons. A threshold of *p* ≤ 0.05 was established to denote statistical significance, while a *p*-value between 0.05 and 0.10 signaled a potential trend warranting further discussion.

## 3. Results

### 3.1. Sequencing Information and Diversity of the Ruminal Microbiota

Our analysis yielded an impressive 2,078,480 clean reads from the rumen fluid samples (see Appendix A). The Good’s coverage indices for the four groups studied were all above 99.8%, indicating that our bacterial identification was extensive and that the sequencing depth was adequately profound for a rigorous analysis of the community. Furthermore, the rarefaction curves approached a plateau (refer to Appendix A), suggesting that the sampling size from each group was adequate. This plateau confirms the thoroughness of our microbial community sampling, as these curves give an estimation of how completely the microbial population has been surveyed.

The alpha diversity index results are presented in Table 1. Regarding community diversity, we observed a trend for SB supplementation’s influence on the Shannon index (*p* = 0.054) and the Pd index (*p* = 0.053). The Simpson index exhibited a quadratic increase with increasing SB supplementation levels (*p* = 0.080), with the MSB group registering a higher Simpson index than the LSB group (*p* < 0.05). The Simpson index was employed as a key metric due to its sensitivity to changes in species dominance and richness. It is essential to understand that higher Simpson index values are indicative of increased dominance by one or a few species within the community, leading to reduced overall diversity. Conversely, lower values suggest a more equitable distribution of species and a richer biodiversity. Therefore, our finding signified that there was a less pronounced dominance by any single species within the LSB group’s microbial community, reflecting a more complex and diverse microbial ecosystem. Therefore, the LSB group demonstrated a more favorable effect on community diversity. In terms of community richness, although the ANOVA *p*-value for the Sobs index was 0.024, the subsequent Tukey–Kramer test yielded a value of 0.054. Moreover, SB supplementation elevated the Ace and Chao1 indices (*p* < 0.05), with the LSB group displaying higher values than the CON group for both indices (*p* < 0.05). As for community evenness, the Shannoneven index increased quadratically with rising SB supplementation levels (*p* < 0.05), while the Simpsoneven index exhibited a linear decrease with increasing SB supplementation levels (*p* < 0.05).

Figure 1A features a Venn diagram, a highly informative graphical tool for displaying the overlap between different sets—in this case, the microbial communities found in the rumen microbiota of preweaning calves across four distinct treatment groups, differentiated by their dosages of SB: the control (CON) group and the low (LSB), medium (MSB), and high (HSB) SB groups. The Venn diagram effectively demonstrates the underlying diversity and richness within these microbial communities by showing how many unique OTUs are present in each group. OTUs are a taxonomic marker used in microbiology to classify groups of closely related individuals, and they provide a way to quantify the presence of different kinds of microorganisms in a sample. The comparative abundance of OTUs in the groups administered with SB (LSB, MSB, and HSB) versus the control group indicates a larger overall microbial population in the treated groups. This suggests the treatments may be nurturing a more complex ecosystem within the calves’ rumen, which could have subsequent effects on their health and development.

The diagram also highlights the extent of shared and unique diversity at the genus level within these microbial communities. Notably, 194 genera, making up 56.89% of the total identified, are common across all four groups, displaying a core microbiome component that is consistent within the ruminal environment of these calves. However, beyond this shared microbial baseline, each of the treatment groups has a distinct set of microbial taxa, with 17 unique genera (4.99%) found in the CON group, 23 (6.74%) in the LSB group, 13 (3.81%) in the MSB group, and 10 (2.93%) in the HSB group. This indicates that while there is a common core of microbial genera, each treatment is associated with a distinct microbial signature, where the LSB group especially stands out with a higher level of genus-specific organisms. Interestingly, despite a larger number of unique OTUs, the HSB group actually shows decreased diversity in terms of the number of phyla and genera compared to the CON group. This could suggest that a higher dosage of SB might lead to a selective environment that supports a high number of unique but more specialized organisms, potentially sacrificing broader diversity at the higher taxonomic levels.

The NMDS plot in Figure 1B provides a graphical interpretation of the differences between the microbial communities across the different groups. It is crucial to understand that beta diversity takes into account the composition of microbial communities and how they differ from one another. The NMDS plot employs weighted UniFrac distances to gauge these differences, focusing on the shared genetic history of the microorganisms present. In essence, the weighted UniFrac distance measures community dissimilarity based on both the presence/absence and the abundance of lineages or branches in a phylogenetic tree, capturing the extent to which groups share a common microbial ancestry.

The stress value, in this case, confirms the validity of the NMDS analysis, falling well below the 0.20 threshold and indicating a good representation of the complex multidimensional data in a more interpretable two-dimensional space. The NMDS plot suggests that the HSB group’s microbial community composition is more divergent from the CON group than the LSB and MSB groups are, hinting at a more pronounced altering effect of high SB levels on the intestinal microbiota. This shift in microbial structure may have meaningful ramifications for the nutrient processing, immunity, and overall health of the calves. Readers need to note that while the *p*-value of 0.067 does not reach the conventional threshold for statistical significance (*p* < 0.05), it is still indicative of a trend that merits attention. It suggests there might be a difference worth exploring with more data or further analysis. The modest R^2^ value, showing the proportion of variation explained by the NMDS, points to other factors also shaping the microbial communities that are not captured in this analysis.

Based on the results from the analysis of the diversity of the ruminal microbiota, SB supplementation at low levels enhances microbial diversity and richness, while high levels may lead to fewer microbial groups. Low SB supplementation (LSB) fosters a more varied and complex microbial ecosystem, with more unique microorganisms and even species distribution. High SB levels (HSB) show the opposite effect. Beta diversity analyses further support the distinct microbial communities among different supplementation levels, with LSB showing beneficial outcomes compared to HSB. Overall, low SB supplementation is key for maintaining a healthy and diverse microbial community.

### 3.2. Profiling the Ruminal Microbial Composition

A taxonomic classification of the ruminal microbiota (stated as relative abundance) was conducted at the phylum and genus levels and is depicted in Appendix A. At the phylum level, the top five phyla in the CON group were *Firmicutes* (46.92%), *Bacteroidota* (27.42%), *Actinobacteriota* (21.89%), *Spirochaetota* (1.20%), and *Synergistota* (0.89%); in the LSB group, the top five phyla were *Firmicutes* (45.49%), *Bacteroidota* (32.26%), *Actinobacteriota* (15.18%), *Spirochaetota* (2.29%), and *Synergistota* (1.50%); in the MSB group, the top five phyla were *Firmicutes* (50.67%), *Bacteroidota* (26.36%), *Actinobacteriota* (19.30%), *Patescibacteria* (1.31%), and *Synergistota* (0.85%); and in the HSB group, the top five phyla were *Firmicutes* (46.97%), *Bacteroidota* (36.72%), *Actinobacteriota* (11.19%), *Spirochaetota* (1.65%), and *Patescibacteria* (1.17%).

At the genus level, the top five genera in the CON group were *Prevotella* (20.18%), *Olsenella* (19.83%), *Succiniclasticum* (5.08%), *Acetitomaculum* (3.86%), and *Lachnospiraceae_NK3A20_group* (3.81%); in the LSB group, the top five genera were *Prevotella* (22.00%), *Olsenella* (13.31%), *Succiniclasticum* (6.19%), *Rikenellaceae_RC9_gut_group* (3.39%), and *Lachnospiraceae_NK3A20_group* (3.27%); in the MSB group, the top five genera were *Prevotella* (17.76%), *Olsenella* (17.54%), *Succiniclasticum* (7.58%), *Acetitomaculum* (3.97%), and *Shuttleworthia* (3.93%); and in the HSB group, the top five genera were *Prevotella* (29.70%), *Olsenella* (10.14%), *Succiniclasticum* (4.86%), *Shuttleworthia* (4.41%), and *Lachnospiraceae_NK3A20_group* (3.79%).

The findings of the statistical analysis regarding the ruminal bacterial phyla and genera are illustrated in Table 2 (showcasing the results of differences) and Appendix A (displaying all results of the statistical investigation). At the phylum level, a linear decrease was observed in the *Actinobacteriota* phylum with growing levels of supplementation (*p* < 0.05). At the genus level, the *Sharpea* and *Lachnospiraceae_FE2018_group* genera exhibited a quadratic effect with the increase in SB supplementation level (*p* < 0.05). The *Oscillospiraceae_UCG-002* and *Pseudoramibacter* genera, on the other hand, were influenced linearly by the increasing SB supplementation level (*p* < 0.05). The HSB group was found to have a higher relative abundance than the other three groups in the *Oscillospiraceae_UCG-002* genera (*p* < 0.05).

Based on the analysis results from the composition of the ruminal microbiota, our study’s examination of the ruminal microbiota at both phylum and genus levels reveals a nuanced response to graded levels of SB supplementation. At the phylum level, *Firmicutes* and *Bacteroidota* were predominant across all groups, but a notable linear decrease in the *Actinobacteriota* phylum was observed with increased supplementation. At the genus level, the diversity is marked by varying responses; while genera such as *Sharpea* and *Lachnospiraceae_FE2018_group* showed a quadratic effect with SB levels, *Oscillospiraceae_UCG-002* and *Pseudoramibacter* displayed a linear correlation with increasing supplementation. Significantly, the HSB group exhibited a higher relative abundance of the *Oscillospiraceae_UCG-002* genus, suggesting a dose-dependent microbial modulation. Collectively, these findings highlight the importance of SB dosage in influencing the ruminal microbial ecosystem.

### 3.3. Sequencing Information and Diversity of the Intestinal Microbiota

From an analysis of forty rectal fecal samples from preweaning calves, we acquired a robust dataset of 1,981,008 clean reads, as outlined in Appendix A. The sufficiency of our sequencing efforts is evidenced by Good’s coverage indices—all exceeding 99.4%. This high coverage index means that we have captured the vast majority of the bacterial species present in these samples, ensuring that our dataset is comprehensive enough for an in-depth analysis of the microbial community.

The rarefaction curves, which assess the richness of species as a function of the total number of individual samples analyzed, reached a plateau, as depicted in Appendix A. This indicates that sampling was sufficient, and further increases in sample size would likely not lead to the discovery of many new species within each group. Thus, we can be confident in the representativeness of our sampling and the reliability of the subsequent community evaluations.

Diving into the microbial communities themselves, Table 3 presents the alpha diversity indices, which measure the variety and abundance of species within a single sample (or group). Although the effects of SB supplementation on the Shannon index—a metric of community diversity—approached statistical significance (*p* = 0.062) and influenced the Simpson index—another community diversity metric in which higher values indicate less diversity—there is a discernible trend that as SB supplementation increases, community diversity may decrease, despite the *p*-value being just outside the traditional threshold for significance (*p* = 0.090). Similarly, the Ace index—which gauges the richness or number of different species present—suggests a potential impact from SB supplementation (*p* = 0.097), but no firm conclusions could be drawn for the observed species (Sobs index) nor the Chao1 index, both measures of species richness, due to their *p*-values being greater than 0.05.

Community evenness—which looks at the relative abundance distributions of the species—also revealed a significant trend. As the level of SB supplementation rose, the Shannoneven index decreased linearly (*p* = 0.045), and the MSB group specifically demonstrated lower evenness compared to the CON group (*p* < 0.05). Likewise, the Simpsoneven index, which adjusts the Simpson index for sample size, displayed a consistent downward trend with increasing SB levels (*p* = 0.030), indicating declines in evenness with higher SB supplementation.

The LSB group was of particular interest as per Figure 2A’s Venn diagram analysis, which disclosed a greater count of OTUs and microorganisms at the phylum, genus, and OTU levels relative to the CON, MSB, and HSB groups. To illustrate, at the genus level, there were 180 genera shared across all groups, constituting 56.60% of the microbial population. The unique genera counts for the CON, LSB, MSB, and HSB groups were 13 (4.09%), 58 (18.24%), 7 (2.20%), and 5 (1.57%), respectively, highlighting the LSB group’s notably higher uniqueness in microbial composition.

Further examining the beta diversity—through the NMDS plot based on the weighted UniFrac distance and the Adonis test—highlights similarities across the microbial communities from each group. The stress value of 0.163 and the R^2^ of 0.097 (*p* = 0.101 as shown in Figure 2B) suggest that despite the differences hinted at by other indices, the overall composition of intestinal microbes is relatively similar among these groups, or at least not starkly different as per the specific beta diversity measures chosen.

Based on the analysis results on the diversity of the intestinal microbiota, the alpha diversity indices indicated a trend toward reduced community diversity and evenness with increased SB supplementation, particularly reflected by an increase in the Simpson index and a decrease in the Shannoneven and Simpsoneven indices; these tendencies were not statistically significant for the Shannon index, the Pd index, the Sobs index, or the Chao1 index. The Venn diagram and beta diversity analysis suggest that the LSB group harbored a slightly richer community at various taxonomic levels without significant differences in overall community composition among the groups. These findings imply that while increasing SB supplementation levels correlate with certain trends in microbial community structure, the impact on overall intestinal microbial diversity may be relatively subtle.

### 3.4. Characterizing the Intestinal Microbial Composition

A comprehensive taxonomic analysis of the intestinal microbiota was carried out, with relative abundance data presented at both the phylum and genus levels, as illustrated in Appendix A.

At the phylum level, the five most abundant phyla in the CON group included *Firmicutes* (61.02%), *Bacteroidota* (35.32%), *Actinobacteriota* (1.07%), *Proteobacteria* (0.65%), and *Spirochaetota* (0.52%); in the LSB group, the five most abundant phyla were *Firmicutes* (63.21%), *Bacteroidota* (29.42%), *Proteobacteria* (3.20%), *Actinobacteriota* (3.15%), and *Spirochaetota* (0.41%); in the MSB group, the five most abundant phyla were *Firmicutes* (54.19%), *Bacteroidota* (38.28%), *Actinobacteriota* (3.90%), *Patescibacteria* (1.39%), and *Proteobacteria* (0.81%); and in the HSB group, the five most abundant phyla were *Firmicutes* (53.11%), *Bacteroidota* (39.99%), *Actinobacteriota* (3.85%), *Patescibacteria* (2.61%), and *Proteobacteria* (0.83%).

At the level of genus classification, within the CON group, the five most predominant genera identified were *Bacteroides* (13.00%), *Lactobacillus* (9.96%), *UCG-005* (6.20%), *Blautia* (6.04%), and *Faecalibacterium* (5.22%); in the LSB group, the five most predominant genera were *Lactobacillus* (11.44%), *UCG-005* (9.83%), *Bacteroides* (7.63%), *Blautia* (6.31%), and *Rikenellaceae_RC9_gut_group* (3.26%); in the MSB group, the five most predominant genera were *UCG-005* (7.87%), *Rikenellaceae_RC9_gut_group* (7.41%), *Lactobacillus* (6.96%), *Bacteroides* (5.66%), and *Blautia* (4.65%); and in the HSB group, the five most predominant genera were *Bacteroides* (12.62%), *UCG-005* (7.77%), *Rikenellaceae_RC9_gut_group* (7.29%), *Lactobacillus* (5.41%), and *Blautia* (4.95%).

The outcomes of the statistical analysis for intestinal bacterial phyla and genera are presented in Table 4 (differential results) and Appendix A (comprehensive statistical analysis results). At the phylum level, there were no significant differences in relative abundance among the four groups for bacterial phyla (*p* > 0.05). At the genus level, *Bacteroides* and *Faecalibacterium* genera displayed a quadratic response to increasing levels of SB supplementation (*p* < 0.05). Both the *Rikenellaceae_RC9_gut_group* and the *Odoribacter* genera exhibited a linear increase in accordance with the escalating SB supplementation level (*p* < 0.05). The MSB and HSB groups demonstrated higher relative abundance compared to the LSB group for the *Rikenellaceae_RC9_gut_group* genera (*p* < 0.05). Lastly, the LSB and MSB groups showed lower relative abundance than the CON group for the *Faecalibacterium* genera (*p* < 0.05).

Upon analyzing the composition of intestinal microbiota, our detailed taxonomic examination identified the prevalent phyla and genera within our study groups’ intestinal flora. *Firmicutes* and *Bacteroidota* emerged as the most abundant phyla across the board. Interestingly, no significant differences at the phylum level were observed among the four groups. However, genus-level variations were notably more distinct. The genera *Bacteroides* and *Faecalibacterium* displayed a quadratic response to SB supplementation, suggesting their relative abundances were influenced by the addition of SB, albeit in a non-linear fashion. Additionally, the genera *Rikenellaceae_RC9_gut_group* and *Odoribacter* demonstrated a definitive linear uptick in proportion to increasing SB supplementation levels. These nuanced changes at the genus level highlight the selective effects of SB supplementation on particular intestinal bacterial communities, emphasizing the complex interplay between dietary supplements and microbiota composition.

### 3.5. The Relationship between the Ruminal Microbiota and the Intestinal Microbiota

A correlation analysis was conducted to examine the relationship between ruminal and intestinal microbiota within each group using a Two-Matrix Correlation Heatmap (Pearson) to assess the alpha diversity index of the microbiota, as depicted in Figure 3. Detailed information (r- and *p*-values) regarding the correlations can be found in Appendix A. Interestingly, calves in the control group (CON) with no SB supplementation displayed predominantly negative correlations between ruminal and intestinal microbiotas (Figure 3A). For instance, the ruminal Pd index was negatively correlated (r = −0.648; *p* = 0.043) with the intestinal Pd index.

Considering the practical implications of the Simpson index (i.e., higher values indicate lower community diversity), the LSB group exhibited positive correlations for all relationships between ruminal and intestinal microbiota. For example, the ruminal Shannon index had a positive correlation (r = 0.828; *p* = 0.003) with the intestinal Shannon index. Moreover, an intriguing result in the LSB group (see Figure 3B and Appendix A) revealed that these relationships revolved around community evenness but not richness-related indices (Shannon and Simpson indices represent community diversity, which reflects the abundance and evenness of species within an ecosystem). Additionally, the majority of correlation coefficients (r-values) in the LSB group demonstrated larger absolute values compared to other groups (Figure 3B).

In the MSB group, relationships between ruminal and intestinal microbiota pertaining to community evenness were also observed. The ruminal Simpsoneven index exhibited a positive correlation (r = 0.700; *p* = 0.024) with the intestinal Simpsoneven index (Figure 3C). However, no discernible relationships between ruminal and intestinal microbiota were detected in the HSB group (Figure 3D, *p* > 0.05).

In conclusion, the data illustrate that SB supplementation exerts a varying influence on the correlation between ruminal and intestinal microbiota among different study groups. For the control group, where SB was absent, there tended to be a negative correlation within the microbial ecosystem. However, with the introduction of SB, particularly within the group with lower SB (LSB), a shift toward positive correlations was notable, especially concerning community evenness as opposed to richness—hinting at SB’s potential role in enhancing microbial harmony in the gut.

The most pronounced positive associations were detected in the LSB group, where higher absolute correlation coefficients signaled a stronger intermicrobial relationship. This promising trend, though, was not observed within the group receiving the highest SB supplementation (HSB), where the link between the ruminal and intestinal microbiota appeared insignificant.

Such results indicate that moderate SB supplementation levels might be integral to fostering a more balanced and uniform gut microbiota composition. This equilibrium in microbial distribution carries significant potential benefits for gut health and the overarching productivity of the animal.

## 4. Discussion

Previous studies have delved into the impact of exogenous butyrate supplements on the ruminal and intestinal microbiota, typically applied in milk replacer scenarios [9,12,13]. In line with these investigations, our recent publication [11] focused on the near-newborn stage, more precisely day 17, while examining the same parameters. Furthermore, our past research established that the optimal exogenous SB supplementation dose seemed to diminish when analyzing the calf growth phenotype, specifically the weekly average daily gain in body weight [11]. Building upon these prior endeavors, the present study aims to extend the scope of examination to a later life stage of dairy calves, specifically day 45.

The current study revealed that exogenous SB supplementation, within a low to high range, impacted the diversity of ruminal microbiota, particularly community richness; this influence even extended to the phylum level of taxonomic analysis. These findings are especially intriguing since milk-supplemented SB bypasses the rumen due to the reflexive sealing of the reticular (or esophageal) groove during calf suckling [21]. Similar results regarding the effects of milk replacer supplemented with SB on ruminal microbiota have been previously reported in dairy calves receiving milk replacer with an encapsulated form of SB [9]. In this study, low SB supplementation (the LSB group) increased community richness (Ace and Chao1 indices) and numerically evened community distribution (Shannoneven and Simpsoneven indices) compared to the control group. This result is consistent with the observation of more bacterial genera at the genus level and more OTUs at the OTU levels. High species richness (the number of different species) and evenness (a balanced distribution of species’ relative abundance) in a community can augment functional redundancy [22]. The role of complementarity and selection mechanisms in shaping the relationship between species richness and some aspects of ecosystem functioning merits future examination [23]. Ultimately, our findings suggest that low SB supplementation benefits preweaning calves by enhancing richness and potentially increasing their adaptability to external challenges, such as the weaning process [12].

A previous study involving a milk replacer administered with a higher SB dose (up to an estimated 24 g/d) for a longer duration (56 days) demonstrated a notable decrease in species richness (measured via the Chao1 index) but no change in diversity (assessed using the Shannon index) within the ruminal microbiota [9]. Parallel outcomes were recorded in the present study; the group with high SB (HSB) demonstrated a notable variance from the control (CON) group within the beta diversity plot. Similar results were observed in a recent study [12], which found that a high level of SB supplementation (15, 30, 45 g/d) via milk replacer failed to increase any index related to ruminal microbiota diversity in 42-day-old preweaning dairy calves. We previously established that the optimal dose of exogenous SB supplementation decreased when evaluating the host’s growth phenotype, specifically the average weekly weight gain of the calves [11]. Additionally, in the present, the *Actinobacteriota* phylum displayed a linear response to increasing levels of SB supplementation, which might correspond with the average weekly weight gain of the calves in our previous study [11], since this *Actinobacteriota* phylum in the rumen has been reported as being positively correlated with body weight in yak calves [24]. In light of these findings, we propose that high doses of exogenous SB supplementation may offer no benefits or could potentially have an inhibitory impact on the development of ruminal microbiota.

As for the composition of the ruminal microbiota for the four genera (*Sharpea, Oscillospiraceae_UCG-002*, *Lachnospiraceae_FE2018_group*, and *Pseudoramibacter*) affected by SB supplementation, the overall results revealed that an overdose of exogenous SB supplementation might cause the relative abundance of most of these potential beneficial bacteria to decrease in the rumen of calves. The *Sharpea* genus is acknowledged for its role in lactic acid production and utilization within the rumen. These bacteria have the crucial capability to degrade lactic acid, thereby preventing a rapid decrease in ruminal pH, which is essential in averting ruminal acidosis—a common metabolic disorder in the rumen [25,26]. This function of *Sharpea* underscores the importance of microbial balance and lactic acid metabolism in maintaining rumen health and overall digestive efficiency. The *Lachnospiraceae_FE2018_group*’s presence in the rumen has been linked to an enhanced fermentation pattern, characterized by a higher acetate/propionate ratio and improved feeding efficiency in Cashmere goats, according to recent findings [27]. This group’s contribution to the literature on gut microbiota composition is not only pivotal for fermentation processes but also seems to foster greater productive performance through the modulation of immune functions, as suggested by the same research. The *Lachnospiraceae* family, known for its ability to break down cellulose and polysaccharides, is instrumental in producing SCFAs such as butyrate [28,29]. These fatty acids are not only vital sources of energy but also play a crucial role in maintaining gut health and immunity. Considering the evidence presented across these studies, it becomes clear that the *Lachnospiraceae_FE2018_group* holds a crucial position in the symbiotic relationship between host and microbiota. It stands at the crossroads of energy homeostasis, immune regulation, and overall feeding efficacy, making it an integral player in the nutritional dynamics of ruminant digestion. Information on the genus *Pseudoramibacter* is relatively sparse. And in the rumen, SCFA-producing bacteria have been reported in cattle [30,31]. Collectively, all these studies provide evidence that the SCFA-producing bacteria in the present study might have been inhibited with the increasing SB supplementation level.

In this study, *UCG-002* in the composition of the ruminal microbiota is classified under the family *Oscillospiraceae*, which is part of the order *Clostridiales*. The genus *UCG-002*, also referred to as unclassified *UCG-002*, represents a group of bacteria from the *Oscillospiraceae* family that are yet to be fully described or named. The role and function of the Oscillospiraceae_UCG-002 group have garnered increasing scientific interest, though their contributions are not yet fully understood. In the rumen of ruminant animals, members of the *Oscillospiraceae* family are generally associated with cellulose degradation, as indicated in previous research [32]. Building on the understanding of other members within the family, it can be postulated that the *UCG-002* genus may also play a role in the metabolism of cellulose and other complex polysaccharides within the rumen environment [33,34]. As such, *Oscillospiraceae_UCG-002* could provide the host with essential nutrients, particularly SCFAs such as propionate or butyrate, which are crucial sources of energy for ruminant animals. Additionally, these SCFAs are implicated in maintaining ruminal pH balance, stimulating the growth of the rumen wall, and modulating the host’s immune functions [4]. Interestingly, the abundance of such beneficial bacteria increases under stress conditions in the host, as evidenced by the significant enrichment observed in the guts of hens at higher densities [35]. This observation may corroborate the findings of our study, which indicate that the abundance of *Oscillospiraceae_UCG-002* significantly increases with elevated levels of SB supplementation. Specifically, in the group with high SB (HSB), there was an approximately six-fold increase in the abundance of *Oscillospiraceae_UCG-002* relative to the control group.

Our current findings in intestinal microbiota reveal that exogenous SB supplementation appears to have limited, or potentially even inhibitory, effects on the microbiota of dairy calves nearing an older age stage (day 45) compared to the newborn period (day 17). This is exemplified through negligible changes in most alpha diversity and beta diversity indices. Furthermore, little change was observed at the phylum level compared to the ruminal microbiota present on day 45. In a related study, we found that the optimal SB supplementation decreased linearly with an increase in the calves’ age, hinting at the potential redundancy of exogenous SB supplementation around day 45 [11]. This evidence supports our present study’s general findings in intestinal microbiota, particularly in the MSB and HSB groups. These groups exhibited an inhibitory effect demonstrated by decreased community evenness, fewer bacterial genera at the genus level, and fewer OTUs at the OTU level.

Regarding the composition of the intestinal microbiota, it is intriguing to note that beneficial bacterial genera impacted by SB supplementation (including *Bacteroides*, *Rikenellaceae_RC9_gut_group*, *Odoribacter*, and *Faecalibacterium*) demonstrated linear or quadratic trends in their abundance with increasing SB supplementation levels. This suggests a potential “selective pressure” or “selective effect” exerted on the beneficial bacteria in the dairy calves’ intestines with varying SB supplementation. These bacterial genera are recognized for their roles in SCFA production or in supporting the host’s digestive functions and nutrient absorption. Of note, the *Rikenellaceae_RC9_gut_group* has been reported to be able to digest fiber, exhibiting a positive correlation with the growth of calves or steers [11,36]. Members of the *Bacteroides* genus, prevalent anaerobic gut bacteria, are pivotal in carbohydrate and protein metabolism and play a vital role in the production of SCFAs [37,38], which are integral for energy and modulating immune and inflammatory responses [39]. The genus *Faecalibacterium*, especially *Faecalibacterium prausnitzii*, comprises a considerable portion of the human colon and is renowned for its production of anti-inflammatory SCFAs, such as butyrate [40,41]. This genus is crucial in maintaining intestinal wall integrity and preventing inflammation and other diseases, thereby promoting general health [42]; these observations have been mirrored in calf research [43]. Extant studies on the *Bacteroides* and *Faecalibacterium* genera support the observation that populations of anti-inflammatory SCFA producers declined with increasing SB supplementation in our study.

Some species within the *Odoribacter* genus share similar functions with *Bacteroides* in the gastrointestinal tract, contributing to the production of SCFAs, notably butyrate, which benefits gut epithelial health and aids in the modulation of immune responses [44,45]. SB has been shown to mitigate lipopolysaccharide-induced inflammation by reducing intestinal damage and upregulating the *Odoribacter* genus in rats [46]. Our findings indicate a linear increase in the abundance of the *Odoribacter* in calves with escalating SB supplementation levels, hinting at possible inflammation with excessive supplementation of exogenous butyrate.

In the current study, the low SB supplementation level (LSB group) seems to neither directly enter the rumen to affect the ruminal microbiota nor reach the distal hindgut and directly impact the hindgut microbiota. One reason for this is the rapid absorption of butyrate by the gut mucosa in the small intestine [47], and another is the increased length of the gastrointestinal tract as animals age compared to the newborn period. Furthermore, the abundance of *Odoribacter* in the intestine increased linearly with the increasing SB supplementation level in this study, a result consistent with the positive correlation between the administration of a milk replacer supplemented with SB and intestinal *Odoribacter* abundance at day 42 in dairy calves [12]. This suggests that the microbiota is highly sensitive to the addition of external SB, with the supplementation of SB in milk exerting a spatially transcendent impact on the hindgut microbiota. It is known that a negative relationship exists between the development of the rumen and the lower gut. A reported inverse relationship exists between the length of the jejunum and the weight of the reticulorumen in calves prior to weaning [48]. There might also be a negative relationship between SB supplementation and the total SCFAs content in the rumen and colon [9]. In our study, this negative correlation was observed in the control group, which exhibited a negative relationship between ruminal microbiota and intestinal microbiota in terms of the Pd index. Interestingly, a positive relationship was observed in the LSB group; for instance, the ruminal microbiota’s Shannon index positively correlated with the intestinal microbiota’s Shannon index. This outcome was not detected in the group with high SB supplementation level (HSB group), suggesting that low SB supplementation can reverse the traditionally observed contradictory development of the rumen and intestines.

Our research suggests that low-level SB supplementation (LSB group) may foster the simultaneous development of both the rumen and intestines by influencing the composition and architecture of the microbiome. This discovery leads to a profound understanding of how LSB could enhance the health and functionality of these digestive components in ruminants, corroborating our previous observation that the optimal level of SB supplementation diminished as calves aged [11]. Previous studies have reported similar findings, where the inclusion of SB (non-encapsulated) in the rumen effectively promoted the development of rumen papillae and mucosal growth in the cecum [49]. Our current study observed that a minor dose of butyrate promotes a positive, collaborative interaction or communication between the rumen and rectal microbiomes. Combined with the findings from these two studies, this appears to be a unique function of butyrate.

Drawing on existing knowledge, we posit the following possible explanations for the importance of butyrate. Butyrate, particularly SB in this case, is an SCFA serving as a crucial energy supply for the gut-lining epithelial cells [4]. Its addition to calves’ diets could strengthen epithelial health along the gastrointestinal tract, thereby nurturing a habitat that promotes microbial variety. Such diverse microbiota simplify the breakdown of various dietary components, improve nutrient uptake, and might even bolster the host’s immune defenses [50]. In addition, SB can improve gut pH stability, which is vital for supporting advantageous microbial populations adept at fermenting different dietary fibers [4,51]. This pH modulation may be one mechanism that reduces the competitive development typically observed between the rumen and intestines. Another factor to consider is butyrate’s role in adjusting the gut’s immune reaction. Early microbial colonization significantly shapes immune development [52], and butyrate is implicated in regulating these early immune pathways. Therefore, SB supplementation could promote a balanced microbial ecosystem within various gut regions, aiding in a coordinated development rather than the disjointed maturation previously noted. Furthermore, the influence of SB extends beyond the microbial realm—it also spans to the gut’s physiological responses. Butyrate is known to modulate gut hormone secretion and ramp up mucin production, both fundamental for maintaining gut barrier integrity and influencing the establishment of the gut microbiome [4]. In summary, our study, supported by existing literature, advocates the idea that SB supplementation in preweaning calves may not only improve the development of individual digestive compartments but also lead to a more integrated and synchronous progression of the overall gastrointestinal microbiome. Verifying these results calls for continued research that should not only iterate these findings but also delve into the mechanistic details of how SB affects the gut environment at the molecular level. Consequently, SB supplementation could represent a major opportunity for enhancing ruminant health and productivity from a young age.

The incorporation of low SB levels demonstrated an enhanced, cooperative outcome between the rumen and intestinal microbiomes, which could have various physiological implications. The first among these is nutrient absorption and energy utilization. The rumen microbiome’s health plays a pivotal role in nutrient absorption in ruminants. It ferments dietary cellulose to yield volatile fatty acids (VFAs), such as acetate, propionate, and butyrate, which are crucial for the host’s supply of energy. Low-level SB supplementation might optimize the production and balance of VFAs, thereby increasing the efficiency of the animal’s energy utilization [11]. Another implication is in gastrointestinal tract development. A healthy rumen and intestinal microbiome are essential for the overall development of the digestive system in ruminants, particularly during their early growth stages. SB supplementation may encourage the synchronized development of the rumen and lower gut, which is beneficial for enhancing the digestion process, affecting the maturation and performance of gastrointestinal tissues [4]. Intestinal barrier function is another area affected by SB supplementation. Butyrate serves as one of the primary energy sources for the cells lining the intestinal epithelium, a critical component for maintaining gut barrier function [49,53]. Proper SB supplementation might reinforce the intestinal wall’s integrity, lowering the likelihood of disease and pathogen intrusion. This could potentially ease the weaning process and contribute to more successful transitions for calves.

The current study elucidates the benefits of a low level of SB supplementation in promoting the development of the rumen microbiota, as well as enhancing the symbiotic link between the microbiome of the rumen and intestines. Conversely, a high level of SB supplementation could impose growth pressures on both the ruminal and intestinal microbiota. Indeed, butyrate plays a vital role during the essential developmental windows of neonates, significantly influencing the advancement of the host’s endocrine, metabolic, and immune systems [4,54]. Consequently, it would be intriguing to investigate the long-term outcomes of both low and high levels of SB supplementation on the host’s endocrine, metabolic, and immune systems and even their growth performance.

## 5. Conclusions

In conclusion, SB supplementation was found to influence ruminal microbiota by altering community diversity, impacting one *Actinobacteriota* phylum and affecting four potentially beneficial bacterial genera: *Sharpea*, *Oscillospiraceae_UCG-002*, *Lachnospiraceae_FE2018_group*, and *Pseudoramibacter*. Compared to its pronounced effect on the rumen, the impact of SB on the intestinal microbiome appeared subtle, with minimal alterations in the microbial diversity indices. However, the composition of the microbial community, particularly beneficial bacterial genera associated with SCFA production such as *Bacteroides*, *Rikenellaceae_RC9_gut_group*, *Odoribacter*, and *Faecalibacterium*, was notably influenced. The findings imply a recommendation for low-level SB supplementation in the milk for preweaning dairy calves. This minor inclusion of SB has demonstrated positive effects on the development of ruminal microbiota and may even bolster the linkage between ruminal and intestinal microbial populations. Conversely, evidence indicates that higher doses of SB do not bring forth observable advantages and could potentially have undesired effects on both ruminal and intestinal microbial ecosystems. These findings not only support the previously reported growth performance of dairy calves but also emphasize the benefits of SB supplementation in suckling milk. Furthermore, this study sheds light on the relationship between ruminal and intestinal microbiota development under SB supplementation during the feeding of dairy calves.

## Figures and Tables

**Figure 1 microorganisms-12-00333-f001:**
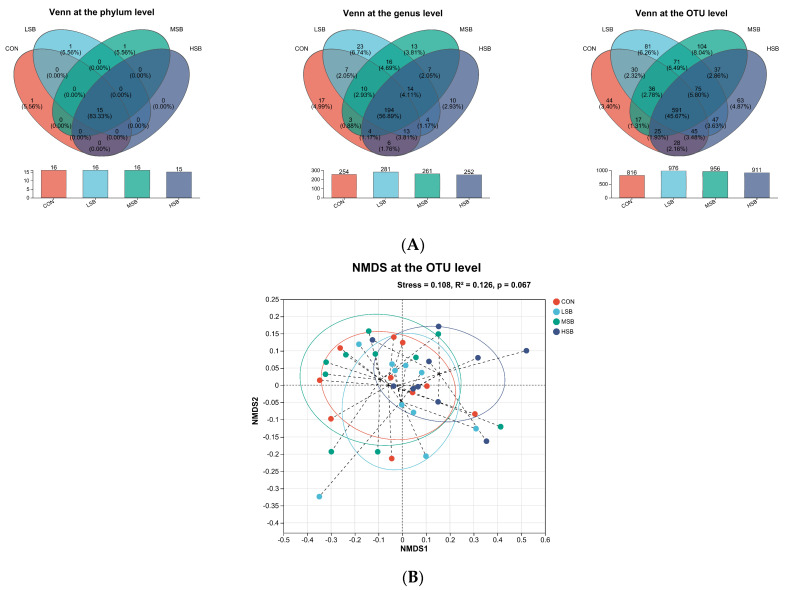
Impact of sodium butyrate in milk on the number of unique microorganisms and beta diversity of ruminal microbiota in preweaning calves. (**A**) The number of unique microorganisms at the phylum, genus, and OTU levels. (**B**) The NMDS plot of beta diversity indices at the OTU level. CON represents the control group, which received milk without sodium butyrate (SB) supplementation. LSB refers to the group with low SB supplementation, MSB refers to the group with medium SB supplementation, and HSB refers to the group with high SB supplementation. Significance was established at *p* < 0.05 with a sample size of 10 for each group.

**Figure 2 microorganisms-12-00333-f002:**
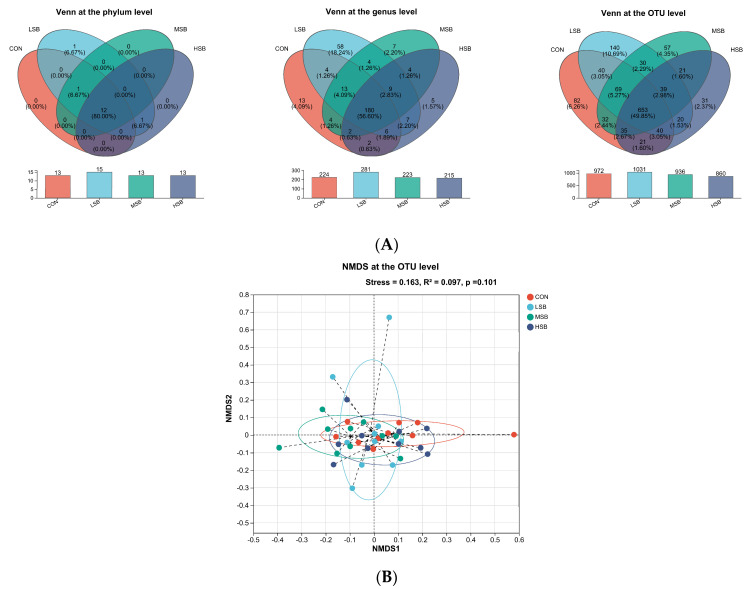
Impact of sodium butyrate in milk on the number of unique microorganisms and beta diversity of intestinal microbiota in preweaning calves. (**A**) The number of unique microorganisms at the phylum, genus, and OTU levels. (**B**) The NMDS plot of beta diversity indices at the OTU level. CON represents the control group, which received milk without sodium butyrate (SB) supplementation. LSB refers to the group with low SB supplementation, MSB refers to the group with medium SB supplementation, and HSB refers to the group with high SB supplementation. Significance was established at *p* < 0.05 with a sample size of 10 for each group.

**Figure 3 microorganisms-12-00333-f003:**
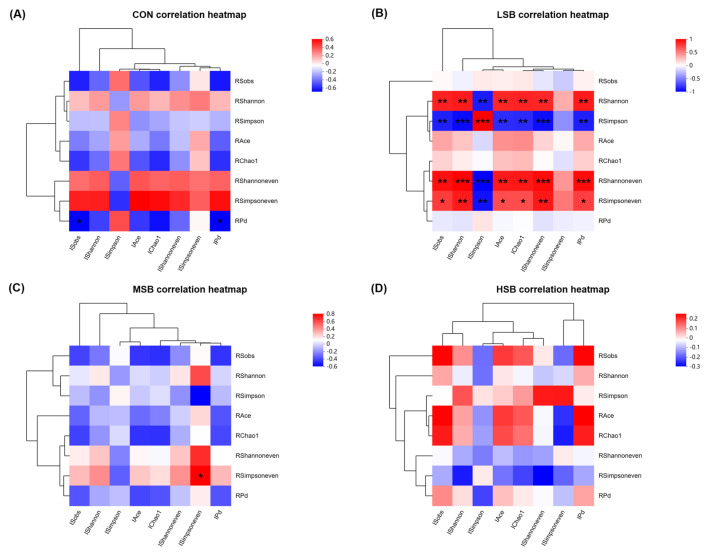
The correlation analysis of the correlation between ruminal and intestinal microbiota in each group using Two-Matrix Correlation Heatmap (Pearson) to analyze the alpha diversity index of microbiota. The correlation heatmaps for each group show the CON group (**A**), LSB group (**B**), MSB group (**C**), and HSB group (**D**), respectively. CON represents the control group, which received milk without sodium butyrate (SB) supplementation. LSB refers to the group with low SB supplementation, MSB refers to the group with medium SB supplementation, and HSB refers to the group with high SB supplementation. Before the diversity index, R represents the ruminal outcome, while I represents the intestinal outcome. Significance was established at *p* < 0.05. 0.01 < * *p* < 0.05, 0.001 < ** *p* < 0.01, and *** *p* < 0.001; *n* = 10.

**Table 1 microorganisms-12-00333-t001:** Impact of sodium butyrate in milk on alpha diversity indices of the ruminal microbiota in preweaning calves.

Items	Supplementation Level, g/d				SEM	*p*-Value		
	0	4.4	8.8	17.6		ANOVA	Linear	Quadratic
Community diversity								
Shannon	3.80	4.03	3.71	4.00	0.049	0.054	0.138	0.113
Simpson	0.0456 ^ab^	0.0362 ^b^	0.0623 ^a^	0.0435 ^ab^	0.0035	0.046	0.064	0.080
Pd	39.58	44.37	41.21	43.41	0.687	0.053	0.386	0.553
Community richness								
Sobs	317.9	371.9	334	373.5	8.08	0.024	0.540	0.854
Ace	404.9 ^b^	501.9 ^a^	436.2 ^ab^	463.9 ^ab^	11.37	0.014	0.258	0.323
Chao1	406.0 ^b^	476.8 ^a^	426.7 ^ab^	462.8 ^ab^	9.96	0.039	0.449	0.625
Community evenness								
Shannoneven	0.660	0.682	0.639	0.676	0.0068	0.111	0.020	0.025
Simpsoneven	0.0694	0.0756	0.0576	0.0656	0.0029	0.174	0.020	0.061

SEM denotes standard error of the mean; *n* = 10. Pd = phylogenetic diversity. Values bearing different superscript letters (a, b) within the same row indicate a significant difference (*p* < 0.05).

**Table 2 microorganisms-12-00333-t002:** Impact of sodium butyrate in milk on the ruminal microbiota (relative abundance, %) at the phylum and genus levels in preweaning calves.

Items	Supplementation Level, g/d				SEM	*p*-Value		
	0	4.4	8.8	17.6		ANOVA	Linear	Quadratic
Phylum level								
*Actinobacteriota*	21.89	15.18	19.30	11.19	1.69	0.117	0.047	0.884
Genus level								
*Sharpea*	3.22	2.01	0.98	2.92	0.387	0.164	0.883	0.029
*Oscillospiraceae_UCG-002*	0.259 ^b^	0.552 ^b^	0.432 ^b^	1.80 ^a^	0.210	0.031	0.007	0.291
*Lachnospiraceae_FE2018_group*	0.509	0.316	0.218	0.332	0.045	0.144	0.214	0.049
*Pseudoramibacter*	0.347	0.255	0.230	0.161	0.026	0.075	0.012	0.538

SEM denotes standard error of the mean; *n* = 10. Values bearing different superscript letters (a, b) within the same row indicate a significant difference (*p* < 0.05).

**Table 3 microorganisms-12-00333-t003:** Impact of sodium butyrate in milk on alpha diversity indices of intestinal microbiota in preweaning calves.

Items	Supplementation Level, g/d				SEM	*p*-Value		
	0	4.4	8.8	17.6		ANOVA	Linear	Quadratic
Community diversity								
Shannon	4.36	4.15	4.07	4.01	0.050	0.062	0.103	0.288
Simpson	0.0274	0.0392	0.0441	0.0463	0.0029	0.090	0.092	0.242
Pd	41.86	41.04	40.11	38.16	0.746	0.350	0.768	0.848
Community richness								
Sobs	432.7	411.4	398.5	369.3	10.80	0.212	0.556	0.969
Ace	535.3	499.4	486.6	446.5	12.76	0.097	0.380	0.833
Chao1	541.0	504.4	496.6	456.1	13.49	0.171	0.480	0.906
Community evenness								
Shannoneven	0.720 ^a^	0.691 ^ab^	0.663 ^b^	0.680 ^ab^	0.0074	0.048	0.045	0.125
Simpsoneven	0.0867	0.0702	0.0589	0.0673	0.0038	0.065	0.030	0.081

SEM denotes standard error of the mean; *n* = 10. Pd = phylogenetic diversity. Values bearing different superscript letters (a, b) within the same row indicate a significant difference (*p* < 0.05).

**Table 4 microorganisms-12-00333-t004:** Impact of sodium butyrate in milk on the intestinal microbiota (relative abundance, %) at the genus level in preweaning calves.

Items	Supplementation Level, g/d				SEM	*p*-Value		
	0	4.4	8.8	17.6		ANOVA	Linear	Quadratic
*Bacteroides*	13.00	7.63	5.66	12.62	1.41	0.169	0.871	0.027
*Rikenellaceae_RC9_gut_group*	4.16 ^ab^	3.26 ^b^	7.41 ^a^	7.29 ^a^	0.670	0.048	0.027	0.666
*Odoribacter*	1.24	0.91	1.27	2.41	0.238	0.121	0.039	0.224
*Faecalibacterium*	5.22 ^a^	2.30 ^b^	3.36 ^b^	4.79 ^ab^	0.415	0.043	0.741	0.015

SEM denotes standard error of the mean; *n* = 10. Values bearing different superscript letters (a, b) within the same row indicate a significant difference (*p* < 0.05).

## Data Availability

The 16S rRNA amplicon sequencing data produced during this research are publicly accessible in the NCBI database, cataloged under the BioProject ID PRJNA904681.

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
