# Peer review of "Modulating Gastrointestinal Microbiota in Preweaning Dairy Calves: Dose-Dependent Effects of Milk-Based Sodium Butyrate Supplementation"

_microorganisms, 2024, doi:10.3390/microorganisms12020333_

Round 1

Reviewer 1 Report

Comments and Suggestions for Authors The manuscript submitted by Ming Xu and the co-authors to the Microorganisms is devoted to the analysis of sodium butyrate supplementation to the calves.   In its current state, the paper first needs a major revision to be considered for publication in Microorganisms. Here are several minor and major issues that the reviewer recommends to resolve:

1) It would be nice to include in the Introduction a diagram showing the connections of butyrate to biochemical pathways physiologically relevant to the calf.

2.1) Specify in the manuscript the reason why such concentrations of SB were chosen as high 17.6 (g/d), medium (8.8 g/d), and low (4.4 g/d). 

  2.2) Why in lines 68-81 are the concentrations of SB indicated first as 4.4 and then 2.2; 8.8 and 4.4 and 17.6 and 8.8 respectively?

3) In lines 82-99 the calves were fed by pasteurized colostrum, which is not physiological. Was any control group that was fed by non-pasteurized colostrum?

4) The general details of 16s rRNA sequencing (library preparation, an instrument used for NGS, etc) should be added to lines 116-130.

5) The discussion of the physiological meaning of the obtained results should be added (in brief) to sections 3.1, 3.2, 3.3, and chapter 4 (in detail).

6) It is a normal situation when such taxonomic ranks as "unclassified_k__norank_d__Bacteria" are generated by the software, but for the journal paper, especially in the thematic journal as Microorganisms, such taxons should be avoided: either resolved or removed from the text and figures (i.e. Fig. 2, Table 2, Fig. 4, Table 5).

7) In the case of section 3.6, what reasons do the authors have to extrapolate the KEGG data to the bacterial taxa that were determined by sequencing a short 16S rRNA fragment? Different strains may or may not have the corresponding genes; the authors do not discuss this assumption in any way. The reviewer believes that this section should be excluded from the article since the results presented in 3.6 are not reasonable in any way in its current state.
8) Section 3.7 lacks a discussion, even a very short one, with references to the literature, about what physiological effect the resulting correlations may have. Comments on the Quality of English Language

In some sections, especially in the introduction, it is difficult for the reader to understand what is being said. 

Author Response

Reviewers 1

Comments and Suggestions for Authors

The manuscript submitted by Ming Xu and the co-authors to the Microorganisms is devoted to the analysis of sodium butyrate supplementation to the calves.   In its current state, the paper first needs a major revision to be considered for publication in Microorganisms. Here are several minor and major issues that the reviewer recommends to resolve:

Overall response:

First and foremost, I wish to extend our deepest appreciation for your thorough understanding and significant regard for our work. We also express our gratitude for your valuable evaluation and suggestions regarding our manuscript.

In response to the insightful assessments and recommendations provided by all reviewers, we have meticulously addressed each comment and made corresponding revisions. All modified sections have been highlighted in green for ease of identification.

Substantial revisions have been made to the manuscript following your guidance, encompassing extensive amendments to the Introduction, Materials and Methods, Results, and Discussion sections. In response to multiple reviewers’ feedback, suggesting an enhanced Introduction and comprehensive discussion of our results, we have increased the number of references from the initial 24 to 60.

Importantly, we have revised the title of our article from the original “Impact of Milk-Based Sodium Butyrate Supplementation on the Microbiota of Preweaning Dairy Calves” to the updated version "Modulating Gastrointestinal Microbiota in Preweaning Dairy Calves: Dose-Dependent Effects of Milk-Based Sodium Butyrate Supplementation.” We hope that our modifications will be met with your approval as well as that of the editorial team.

Regarding the language, we have undertaken professional editing service, as recommended by the Editor, and paid for language polishing (https://www.mdpi.com/authors/english). The certificate of editing is English-Editing-Certificate-76506.

We deeply thank you for reassessing our manuscript, microorganisms-2811650, and look forward to your approval of the changes we have implemented.

Sincerely

Specific comments and responses

1.) It would be nice to include in the Introduction a diagram showing the connections of butyrate to biochemical pathways physiologically relevant to the calf.
Response: Thank you for your valuable suggestion regarding the incorporation of a diagram to demonstrate the connections between butyrate and physiologically relevant biochemical pathways in the calf in the Introduction of our manuscript.

We appreciate the time you’ve taken to review our work and recognize that your recommendation could help readers better visualize and comprehend the significance of butyrate in the context of calf physiology. While it is not standard practice within the journal “Microorganisms” to include diagrams in the Introduction, we understand the potential benefit such a figure could offer.

After careful consideration of your suggestion and review of the structure commonly employed in “Microorganisms,” we believe that including a detailed explanation within the text suffices for the scope and aims of our current manuscript. Nonetheless, we are committed to ensuring the clarity and accessibility of our work. Thus, we propose to expand our textual description in the Introduction to more explicitly outline these biochemical pathways and their relevance. (Lines 39-60)

We trust that this approach will satisfy the needs of the readership, without diverging from the journal’s conventional formatting.

Thank you once again for your insightful comments, which invariably contribute to the enhancement of our manuscript. We hope our rationale is understandable, and we are open to further discussion if required.

2.1) Specify in the manuscript the reason why such concentrations of SB were chosen as high 17.6 (g/d), medium (8.8 g/d), and low (4.4 g/d). 

2.2) Why in lines 68-81 are the concentrations of SB indicated first as 4.4 and then 2.2; 8.8 and 4.4 and 17.6 and 8.8 respectively?
Response:

Response to Comment 2.1:

We appreciate the request for clarification regarding the chosen concentrations of SB. In our previous study, which we inadvertently did not mention in the current manuscript, we established the rationale behind the selected dosages; allow us to elucidate as follows. Two experiments were conducted, beginning with a preexperiment that evaluated calves fed milk containing 0, 2, 4, and 6 g/L of SB supplementation at a daily intake of 8.8 L. We observed that 2 g/L dosage did not elicit any noticeable effects during the preexperiment, while dosages of 4 and 6 g/L resulted in a reduced intake by some calves initially. Consequently, 2 g/L was determined to be the maximum concentration for the subsequent formal experiment, which examined dosages of 0, 0.5, 1, and 2 g/L SB. However, to maintain consistent levels of supplementation as the amount of milk fed to newborn calves increased from lesser to greater volumes, we adjusted the 2 g/L dose to a fixed daily amount, specifically 17.6 g/d. Therefore, in the formal experiment, we ultimately utilized doses of 0, 4.4, 8.8, and 17.6 g/d. Fortuitously, calf growth performance improved with these levels of SB supplementation, and we established an optimal SB supplementation level of approximately 8.78 g/d or roughly 1 g/L of milk across the entire period. Accordingly, our experiment defined 4.4 g/d as the low dose, 8.8 g/d as the medium dose, and 17.6 g/d as the high dose.

For improved clarity, the manuscript has been revised to include the following: The highest level of SB supplementation was set at 17.6 g/d, referenced from prior groundwork established in a pre-experiment and documented by Wu et al., 2023 [13], where we saw no further promotive effects on preweaning calves at levels exceeding this concentration in milk. Hence, 17.6 g/d was selected as the ceiling. To comprehen-sively explore the effects of SB supplementation on calf growth, we established two in-termediary dosages between the zero-supplementation baseline (0 g/d) and the peak (17.6 g/d). These were 4.4 g/d, representing the low level, and 8.8 g/d, representing the moderate level, facilitating an evaluation of the incremental effects across distinct SB concentrations in our study. (Lines 103-111)

Response to Comment 2.2:

The initial presentation of SB supplementation concentrations in the manuscript indeed caused some confusion, which we aim to clarify. We supplemented the calves’ milk with daily doses of SB at 4.4 g for the LSB group, 8.8 g for the MSB group, and 17.6 g for the HSB group. Since the calves were fed twice per day, this effectively made the per feeding doses 2.2 g for LSB, 4.4 g for MSB, and 8.8 g for HSB. We understand now that the manuscript’s original articulation of daily SB supplementation did not consistently reflect this, so we have made the following modification to address the issue: we revised the original text, “The treatments involved milk with varying amounts of SB supplementation for calves: (1) Control without SB (0 g/d), CON; (2) Low level of SB (4.4 g/d), LSB; (3) Medium level of SB (8.8 g/d), MSB; and (4) High level of SB (17.6 g/d), HSB…” to now state, “The treatments involved feeding calves milk with varying amounts of SB supplementa-tion for calves: 1) control without SB, CON; 2) low level of SB, LSB; 3) medium level of SB, MSB; and 4) high level of SB, HSB. The daily supplementation of SB for the LSB, MSB, and HSB groups was established at 4.4 g, 8.8 g, and 17.6 g, respectively, equally divided across two feedings, resulting in per-feeding quantities of 2.2 g for LSB, 4.4 g for MSB, and 8.8 g for HSB.” (Lines 98-103)

3) In lines 82-99 the calves were fed by pasteurized colostrum, which is not physiological. Was any control group that was fed by non-pasteurized colostrum?
Response: Yes, it is common knowledge that colostrum, in its natural state, is typically obtained directly from the cow without undergoing any form of sterilization. In our trial, all the calves were fed pasteurized colostrum, and we did not include a control group that was fed non-pasteurized colostrum. While it is plausible to concern oneself with the potential alterations in the immunological and nutritional components of colostrum due to pasteurization, and the subsequent effects on the representativeness and physiological significance of the experimental outcomes, our approach is consistent with standard practices on farms in China. Furthermore, pasteurization does not negatively impact the acquisition of passive immunity in newborn calves and might even be more beneficial in reducing health risks. For instance, Malik et al. (2022) noted that “Heat treatment of colostrum at 60°C decreases colostrum immunoglobulins but increases serum immunoglobulins and serum total protein.” Additionally, Stabel J. R. (2008) reported that “Pasteurization of colostrum reduces the incidence of paratuberculosis in neonatal dairy calves.”

Despite these considerations, the decision to feed calves pasteurized colostrum in our trials was made to ensure that the newborns were not adversely influenced by bacteria and harmful microbes potentially present in the colostrum. As colostrum can be a carrier of infectious agents, heat treatment of raw colostrum is a precautionary measure employed to eliminate or significantly lower the pathogen load, thereby preventing the influence on gut microbial colonization and overall growth of the neonates.

References:

Malik, M. I., Rashid, M. A., & Raboisson, D. (2022). Heat treatment of colostrum at 60°C decreases colostrum immunoglobulins but increases serum immunoglobulins and serum total protein: A meta-analysis. Journal of dairy science, 105(4), 3453–3467. https://doi.org/10.3168/jds.2021-21231

Rabaza, A., Fraga, M., Mendoza, A., & Giannitti, F. (2023). A meta-analysis of the effects of colostrum heat treatment on colostral viscosity, immunoglobulin G concentration, and the transfer of passive immunity in newborn dairy calves. Journal of dairy science, 106(10), 7203–7219. https://doi.org/10.3168/jds.2022-22555

Stabel J. R. (2008). Pasteurization of colostrum reduces the incidence of paratuberculosis in neonatal dairy calves. Journal of dairy science, 91(9), 3600–3606. https://doi.org/10.3168/jds.2008-1107

Godden, S. M., Smolenski, D. J., Donahue, M., Oakes, J. M., Bey, R., Wells, S., Sreevatsan, S., Stabel, J., & Fetrow, J. (2012). Heat-treated colostrum and reduced morbidity in preweaned dairy calves: results of a randomized trial and examination of mechanisms of effectiveness. Journal of dairy science, 95(7), 4029–4040. https://doi.org/10.3168/jds.2011-5275

4) The general details of 16s rRNA sequencing (library preparation, an instrument used for NGS, etc) should be added to lines 116-130.
Response: Thank you for your constructive feedback on our manuscript. We acknowledge the need for greater detail regarding the 16S rRNA sequencing process, specifically library preparation and the next-generation sequencing (NGS) instrument utilized in our study.

In response to your comments, we have revised Lines 174-190 of our manuscript to include a more comprehensive description of the methods used. The updated text now details the use of the E.Z.N.A.® Soil DNA Kit for DNA extraction and the subsequent quality assessment via 1% agarose gel electrophoresis and NanoDrop 2000 spectrophotometry.

For the amplification of the V3–V4 hypervariable regions, we described the PCR reaction ingredients and conditions, including the specifics of the PrimeSTAR HS DNA Polymerase and thermal cycling protocol. We also elaborated on the steps taken for PCR product purification, which were omitted in our initial submission.

Furthermore, we have specified the use of the Illumina MiSeq sequencing platform, which is particularly relevant for high-accuracy microbial diversity analyses. This addition underscores the reliability of the sequencing data presented in our study.

The text now comprehensively explains the demultiplexing and stringent quality filtering processes, employing tools such as fastp and FLASH, followed by OTU clustering via UPARSE with a similarity threshold of 97%. A more thorough explanation of our use of the RDP Classifier algorithm against the Silva 16S rRNA database has also been included for taxonomic classification, enhancing clarity and reproducibility of our research methods.

We believe that these amendments effectively address your concerns, offering a clear and detailed narrative of our 16S rRNA sequencing approach from library preparation to sequencing data processing. We trust that these changes will significantly improve the manuscript and appreciate your guidance in improving the quality of our work.

Thank you for your attention to this matter, and we look forward to any further suggestions you may have.

5) The discussion of the physiological meaning of the obtained results should be added (in brief) to sections 3.1, 3.2, 3.3, and chapter 4 (in detail).
Response: We are grateful for your insightful comments and reminders. They prompted us to thoroughly revisit the author guidelines of the journal Microorganisms, particularly the requirements for the Results section which are stated as follows:

“3. Results

This section may be divided by subheadings. It should provide a concise and precise description of the experimental results, their interpretation, as well as the experimental conclusions that can be drawn.”

In accordance with this guidance, we have meticulously refined our interpretation of the experimental results in the Results section. Furthermore, it was not possible to discuss the physiological meaning of the obtained results in this section in the manner you described (“discussion of the physiological meaning of the obtained results”), as this goes against the journal’s specific instructions. Instead, we have drawn experimental conclusions for each part of the results, as required by the journal. We kindly ask you to consider the journal’s guidelines and approve these modifications.

As you have rightly suggested, we have thoroughly discussed the physiological significance of the obtained results in the Discussion section. Upon reviewing this section for any potential omissions, we have made the following additions and elaborations:

In the second paragraph of the Discussion (Lines 556-570), we have addressed the physiological implications of higher microbial diversity.

In the third paragraph, we have discussed the reduction in ruminal microbial diversity due to excessive sodium butyrate addition and its physiological meanings, which also correlates with the Actinobacteriota phylum mentioned in the results for 3.2 Composition of the Ruminal Microbiota.

Our discussion on diversity incorporates aspects from 3.3 Diversity of the Intestinal Microbiota.

Regarding 3.2 Composition of the Ruminal Microbiota, we realized we had overlooked descriptions for rumen inhabitants such as Sharpea, Oscillospiraceae_UCG-002, Lachnospiraceae_FE2018_group, and Pseudoramibacter, which we have now addressed in the fourth and fifth paragraphs of the Discussion section.

For 3.4 Composition of the Intestinal Microbiota, we have expanded on Bacteroides, Odoribacter, and Faecalibacterium in the seventh and eighth paragraphs of the Discussion.

With respect to 3.5 The Relationship Between the Rumen and Intestinal Microbiota, we recognized a lack of references discussing the possible reasons and physiological meanings behind the observed relationship, especially in the LSB group. We have now supplemented our discussion on this topic in the eleventh and twelfth paragraphs.

Overall, we have summarized the existing discussions, indicating where they are located, and we have supplemented the discussions on results that were previously not addressed in detail.

We sincerely appreciate your suggestions and hope that our revisions meet your expectations and the journal’s standards.

6) It is a normal situation when such taxonomic ranks as "unclassified_k__norank_d__Bacteria" are generated by the software, but for the journal paper, especially in the thematic journal as Microorganisms, such taxons should be avoided: either resolved or removed from the text and figures (i.e. Fig. 2, Table 2, Fig. 4, Table 5).
Response: Thank you for your insightful comments on our manuscript. We understand your concern regarding the presence of nonspecific taxonomic ranks such as “unclassified_k__norank_d__Bacteria” and “norank” in our paper and the accompanying figures and tables.

Consistent with the high standards expected by the journal ‘Microorganisms’, and in order to enhance the clarity and utility of our findings, we have decided to remove all instances of these unresolved taxa from our text and supporting figures (specifically, Table 2 and Table 5). Additionally, we have observed that Figures 2 and 4 presented content that was redundant with the information displayed in our tables. In conjunction with the complete results delineated in Supplementary Table S2 and Table S4 within the supplemental material, we have determined that retaining Figures 2 and 4 in the main body of our manuscript would be superfluous. Therefore, we have decided to remove Figures 2 and 4 from the manuscript to streamline the presentation and avoid unnecessary repetition. These changes aim to present our data in the most informative and precise way possible, focusing on well-resolved and informative taxonomic classifications that are relevant to our readers.

The necessary revisions have been made to the manuscript (also included the Supplementary Table S2 and Table S4), ensuring that the taxonomic representation in our study is both scientifically rigorous and coherent with established microbial nomenclature. We believe these adjustments will improve the manuscript and adhere to the journal’s thematic focus.

We appreciate your guidance, which has undeniably enhanced the quality of our paper. Please do let us know if there are any further modifications or clarifications you feel would benefit this work.

7) In the case of section 3.6, what reasons do the authors have to extrapolate the KEGG data to the bacterial taxa that were determined by sequencing a short 16S rRNA fragment? Different strains may or may not have the corresponding genes; the authors do not discuss this assumption in any way. The reviewer believes that this section should be excluded from the article since the results presented in 3.6 are not reasonable in any way in its current state.
Response: Thank you for your critical evaluation of our manuscript. We have reconsidered our approach, particularly in sections 3.3 “Function of the Ruminal Microbiota” and 3.6 “Function of the Intestinal Microbiota,” where we attempted to use 16S rRNA gene sequencing data to predict metabolic functions using KEGG.

We understand from your valuable feedback that our methodology for linking the 16S rRNA gene with specific functional profiles may not accurately reflect the genomic and metabolic diversity present within different bacterial strains. Upon close reflection, we agree that such assumptions may lead to overinterpretations of the metabolic capabilities of the identified taxa.

As a response to your concerns, and in an effort to maintain a high level of scientific rigor, we have decided to remove both sections 3.3 and 3.6 from our manuscript. This ensures that we do not present any potentially misleading functional predictions that are not fully supported by our data.

In the revised manuscript, sections 3.3 and 3.6 are now excluded, and we have made corresponding adjustments to the rest of the text to preserve the flow and integrity of the manuscript. Any related conclusions or inferences drawn from these sections have also been revised or removed.

We believe these revisions address the concerns you have raised and improve the reliability and clarity of our study. As always, we are grateful for your guidance and for the opportunity to enhance our work.

8) Section 3.7 lacks a discussion, even a very short one, with references to the literature, about what physiological effect the resulting correlations may have.

Response: Thank your comments. We would like to clarify that our discussion of the physiological impact is included in the manuscript’s 12th paragraph in the Discussion section. Specifically, we elaborate on how the symbiotic relationship between the rumen and intestine microbiomes is enhanced by the supplementation of SB. This supplementation is postulated to contribute to the improved health and performance observed in postpartum Simmental cows. We appreciate the reviewer’s feedback and have ensured that pertinent references to existing literature on the topic have been incorporated to support our discussion.

Please let us know if any further clarification or additional information is required. We are committed to addressing all concerns to improve the quality and clarity of our manuscript. Thank you again for your valuable feedback.

Comments on the Quality of English Language

In some sections, especially in the introduction, it is difficult for the reader to understand what is being said.

Response:

Thank you for your feedback concerning the quality of English language in our manuscript. We acknowledge the importance of clear and comprehensible writing, especially in the introduction where setting the stage for our research is critical.

In response to your concerns, we have rigorously revised our manuscript for language clarity and fluency. We have also enlisted the services of a professional editing service, which specializes in academic writing and adheres to American English standards, to ensure that the language quality meets the high expectations of your journal and its readership. You can find more about the service here: paid editing servicehttps://www.mdpi.com/authors/english. (English-Editing-Certificate-76506)

We trust that these efforts have substantially improved the readability of our text and we are confident that the revised manuscript now communicates our research more effectively.

Reviewer 2 Report

Comments and Suggestions for Authors

The paper “Impact of Milk-Based Sodium Butyrate-Supplementation on Microbiota of Preweaning Dairy Calves” does not present/include enough originality. Please see attached document for details.

Author Response

Reviewers 2

The paper “Impact of Milk-Based Sodium Butyrate-Supplementation on Microbiota of Preweaning Dairy Calves” does not present/include enough originality, specifically:

  • It present the same experimental procedure as described in the “Effects of Sodium Butyrate Supplementation in Milk on the Growth Performance and Intestinal Microbiota of Preweaning Holstein Calves”, Animals, 2023, 13, 2069 manuscript.
  • Using identical concentrations of Sodium Butyrate (SB): 0; 4.4; 8.8 si17.6 g/d
  • The same analytical method was used for samples as presented in the manuscript mentioned above
  • Experimental data was processed using the same statistical method

Overall response:

Dear Reviewer,

We are immensely grateful for your insightful and precise evaluation of our manuscript. It indeed has a relationship with our previously published paper, encompassing the same experimental designs, sample analysis methods, and statistical methodologies.

Our former article focused on the growth performance throughout a six-week trial period and the intestinal microbiota at two weeks of age. This current manuscript is a continuation, presenting the rumen and intestinal microbiota at six weeks. The fundamental question driving our research is the potential for Sodium Butyrate (SB) supplementation to confer consistent benefits to the gastro-intestinal microbiota development as calves age. Counter to the prevailing belief that SB invariably promotes microbial development, our findings suggest a nuanced scenario. The external addition of butyrate—provided through SB—does not demonstrate the same pronounced benefits to the microbial development in older calves as seen in those at two weeks, particularly even not within the medium supplementation group (MSB).

Another one of our key discoveries is that low-dose exogenous butyrate supplementation appears to fortify the connectivity between rumen and intestinal microbiota, an effect not observed under medium or high doses. Therefore, our study indicates that from a microbiological perspective, the addition of low-dose SB may have beneficial effects as the calves grow older—insights not presented in the previous article.

We believe that these novel findings contribute substantial reference value, present intriguing results, and warrant publication in your esteemed journal.

Regarding the use of identical experimental and analytical methods, we are of the conviction that such approaches maximize the revelation of our results and ensure that the data are directly comparable. Moreover, these methods represent the best practices accepted within our field, making their use inevitable for continuity and consistency with our past work.

We kindly ask for your consideration of our explanations. Thank you very much. Should you believe there are areas where our manuscript could be further improved, please provide your invaluable suggestions.

Sincerely

  • Paragraphs 3.2. (Composition of the Ruminal Microbiota) and 3.5. (Composition of the Intestinal Microbiota) present the same data both in textual and diagram formats (figures 2 and 4). Generally, the recommendation is to present data in using a single format, either a table, a diagram or text.

Response: We sincerely appreciate the insightful feedback provided. In response to your astute observations, we have removed Figures 2 and 4 from our manuscript. This action was taken to ensure that our results are not redundantly displayed, as these figures portrayed the same data that can be found in Supplementary Tables S2 and S4. In adherence to your valuable recommendation to present data in a single format, we have opted to exclusively use tables to convey the composition of ruminal and intestinal microbiota.

Additionally, we express our gratitude for the guidance on the use of nonspecific taxonomic ranks. In accordance with Reviewer 1’s suggestion, and in an effort to meet the rigorous standards of the journal ‘Microorganisms’, we have eliminated all instances of “unclassified_k__norank_d__Bacteria” and “norank” from our paper and the accompanying tables. This refinement has been undertaken to increase the clarity and applicability of our findings. Thank you once again for your constructive comments, which have significantly contributed to the enhancement of our manuscript.

Round 2

Reviewer 1 Report

Comments and Suggestions for Authors

The manuscript has been significantly improved according to the reviewers' comments

Author Response

Dear Reviewer,

We sincerely appreciate your overall positive evaluation stating, “The manuscript has been significantly improved according to the reviewers’ comments,” and though you did not have specific suggestions or comments, we still value and have taken into account the areas you noted as “Can be improved,” including:

“Does the introduction provide sufficient background and include all relevant references?”

“Are all the cited references relevant to the research?”

“Is the research design appropriate?”

“Are the methods adequately described?”

“Are the results clearly presented?”

“Are the conclusions supported by the results?”

Regarding your query on whether the introduction provides “sufficient background and includes all relevant references” and whether “all the cited references are pertinent to the research,” we have made substantial revisions to the beginning of the introduction. Notably, we combined what were originally the first and second paragraphs into a singular, cohesive paragraph that now aligns more closely with the subject matter of our study. Additionally, we conducted a thorough review of our cited references, removing those focused on human children and replacing them with literature pertinent to calves. Following a careful examination, we’ve deleted nine references and introduced three new ones, resulting in a reduction from the initial 60 references to 54. Your advice and comments have been invaluable in guiding these amendments.

As for your comment on “Is the research design appropriate?”, we understand the importance of research design, though modifying it at this stage is not feasible. We would deeply appreciate any specific advice or suggestions for improvement that we could apply to future research endeavors.

Confronting your point on “Are the methods adequately described?”, we have expanded our Materials and Methods section (Lines 110-120) to include details on calf-rearing management practices. Additionally, we made sure to clarify screening techniques for microbial cultures (Lines 219-221), expressly noting the removal of non-specific taxonomic ranks and terms such as ‘unclassified’ and ‘norank’ from the results section.

Addressing your observation regarding “Are the results clearly presented?”, we thoroughly reviewed and refined our results section, especially emphasizing the subsection “3.5. The Relationship Between the Ruminal Microbiota and the Intestinal Microbiota”.

In response to “Are the conclusions supported by the results?”, we regret that our initial manuscript may have omitted the full description and synthesis of all results into the conclusions. Thus, we have reworked paragraphs 1 to 4 in the Conclusions section, with the first sentence detailing the overall findings regarding the rumen microbiota, and the subsequent two highlighting the general results for the intestinal microbiota. These enhancements were missing in the previous draft and represent an important addition to our revised submission.

In closing, your feedback has been crucial for refining our research, and we thank you profoundly for your input and time. Your effort and guidance are greatly appreciated.

Warm regards,

Authors

References Change List

We have re-evaluated the references within our manuscript to ensure their pertinence to the study’s content. Consequently, we have removed certain citations, primarily from the first paragraph of the Introduction and the Discussion sections, reducing the number of references from 60 to 54. Below, we provide detailed reasons and specific locations within the manuscript for the removal and replacement of references:

In the Introduction, at Line 32, the following references were removed (Relevance to the human infant gut was considered low for the context of this manuscript, hence removed.):

  1. Milani, C.; et al. The First Microbial Colonizers of the Human Gut: Composition, Activities, and Health Implications of the Infant Gut Microbiota. Microbiology and Molecular Biology Reviews 2017, 81.
  2. Robertson, R.C.; et al. The Human Microbiome and Child Growth - First 1000 Days and Beyond. Trends in Microbiology 2019, 27, 131-147.
  3. Yao, Y.; et al. The Role of Microbiota in Infant Health: From Early Life to Adulthood. Frontiers in Immunology 2021, 12.

Instead, the following more pertinent references were cited:

  1. Arshad, M.A.; et al. Gut Microbiome Colonization and Development in Neonatal Ruminants: Strategies, Prospects, and Opportunities. Animal Nutrition, 2021, 7, 883-895.
  2. Cangiano, L.R.; et al. ADSA Foundation Graduate Student Literature Review: Developmental Adaptations of Immune Function in Calves and the Influence of the Intestinal Microbiota in Health and Disease. Journal of Dairy Science 2023.

Removed reference at Introduction, Line 47, due to irrelevance to the subject of calves:

  1. Bronner, D.N.; et al. Genetic Ablation of Butyrate Utilization Attenuates Gastrointestinal Salmonella Disease. Cell Host & Microbe 2018, 23.

Replaced citation at Line 762 due to low relevancy to the discussion on “butyrate influencing the advancement of the host’s endocrine, metabolic, and immune systems”:

  1. Robertson, R.C.; et al. The Human Microbiome and Child Growth - First 1000 Days and Beyond. Trends in Microbiology 2019, 27.

Replaced with more relevant literature:

  1. Guilloteau, P.; et al. From the Gut to the Peripheral Tissues: The Multiple Effects of Butyrate. Nutrition Research Reviews 2010, 23.
  2. van der Hee, B.; Wells, J.M. Microbial Regulation of Host Physiology by Short-chain Fatty Acids. Trends in Microbiology 2021, 29.

Removal of reference in the Discussion, Lines 571, due to lack of direct relevance to the rumen microbiota:

  1. Villot, C.; et al. Early Supplementation of Saccharomyces cerevisiae Boulardii CNCM I-1079 in Newborn Dairy Calves Increases IgA Production in the Intestine at 1 Week of Age. Journal of Dairy Science 2020, 103.
  2. Rostoll Cangiano, L.; et al. Saccharomyces cerevisiae Boulardii Accelerates Intestinal Microbiota Maturation and is Correlated with Increased Secretory IgA Production in Neonatal Dairy Calves. Frontiers in Microbiology 2023, 14.

Removed references in the Discussion, Lines 677, due to the removal of content unrelated to inflammation and the corresponding references:

  1. Luo, D.; et al. Niacin Protects against Butyrate-Induced Apoptosis in Rumen Epithelial Cells. Oxidative Medicine and Cellular Longevity 2019, 2019.
  2. Wang, F.; et al. Sodium Butyrate Inhibits Migration and Induces AMPK-mTOR Pathway-dependent Autophagy and ROS-mediated Apoptosis via the miR-139-5p/Bmi-1 Axis in Human Bladder Cancer Cells. FASEB Journal 2020, 34.
  3. Yu, C.; et al. Effect of Exercise and Butyrate Supplementation on Microbiota Composition and Lipid Metabolism. The Journal of Endocrinology 2019, 243.

We have ensured that the revised references are more closely aligned with the scope of our study, specifically focusing on the development and function of the gut microbiome in neonatal ruminants and its implications for health and disease. Our amended references list should provide a clearer framework for readers to understand the context and findings of our research.

We believe that these adjustments have strengthened the reference list of our manuscript and ultimately enrich our contribution to the scientific dialogue.

Reviewer 2 Report

Comments and Suggestions for Authors

The manuscript has been improved in terms of description and presentation of data, but as a scientific novelty, no major changes can be seen,

Author Response

Dear Reviewer,

Thank you for your feedback and for acknowledging the enhancements we made concerning the description and presentation of data. Your comment on the scientific novelty of the manuscript prompts us to further clarify the innovative aspects of our research.

Our study’s primary novel contribution involves investigating the effects of sodium butyrate (SB) supplementation in milk on the rumen microbiota, as detailed in sections “3.1. Sequencing Information and Diversity of the Ruminal Microbiota” and “3.2. Profiling the Ruminal Microbial Composition.” Another innovative aspect is how the administration of smaller doses of SB can promote harmonious development between the rumen and intestinal microbiota, illustrated in the results section “3.5. The Relationship Between the Ruminal Microbiota and the Intestinal Microbiota.” These points were indeed presented in previous versions of our manuscript, which is why there may not have been perceived “major changes” in these areas.

However, we have also strengthened the discussion of the influence of SB on the rumen microbiota in paragraphs 3 to 6 of the Discussion section, as well as expanded the dialogue regarding the relationship between the rumen and intestinal microbiota in paragraphs 2 to 5 from the end of the Discussion section in our previous revision.

Furthermore, to underscore the importance of innovation in our work, we have revised the subsection titled “3.5. The Relationship Between the Ruminal Microbiota and the Intestinal Microbiota,” and enhanced the manuscript’s conclusion to better reflect the novel contributions of our study.

Your insights have been invaluable in guiding these revisions and we extend our gratitude for your input that has aided in the advancement of our manuscript.

Warm regards,

Authors